# *In vivo* Polycystin-1 interactome using a novel *Pkd1* knock-in mouse model

**Cheng-Chao Lin**[1], **Luis F. Menezes**[1], **Jiahe Qiu**[1], **Elisabeth Pearson**[1], **Fang Zhou**[1], **Yu Ishimoto**[1], **D. Eric Anderson**[2], **Gregory G. Germino**[1]\*

**1** Polycystic Kidney Disease Section, Kidney Disease Branch, National Institute of Diabetes and Digestive and Kidney Diseases, National Institutes of Health, Bethesda, Maryland, United States of America,
**2** Advanced Mass Spectrometry Core, National Institute of Diabetes and Digestive and Kidney Diseases, National Institutes of Health, Bethesda, Maryland, United States of America

☯ These authors contributed equally to this work.
\* germinogg@mail.nih.gov

**Data Availability Statement:** The mass spectrometry proteomics data have been deposited to the ProteomeXchange Consortium with identifier PXD035250 and 10.6019/PXD035250.

## Abstract

*PKD1* is the most commonly mutated gene causing autosomal dominant polycystic kidney disease (ADPKD). It encodes Polycystin-1 (PC1), a putative membrane protein that undergoes a set of incompletely characterized post-transcriptional cleavage steps and has been reported to localize in multiple subcellular locations, including the primary cilium and mitochondria. However, direct visualization of PC1 and detailed characterization of its binding partners remain challenging. We now report a new mouse model with HA epitopes and eGFP knocked-in frame into the endogenous mouse *Pkd1* gene by CRISPR/Cas9. Using this model, we sought to visualize endogenous PC1-eGFP and performed affinity-purification mass spectrometry (AP-MS) and network analyses. We show that the modified *Pkd1* allele is fully functional but the eGFP-tagged protein cannot be detected without signal amplification by secondary antibodies. Using nanobody-coupled beads and large quantities of tissue, AP-MS identified an *in vivo* PC1 interactome, which is enriched for mitochondrial proteins and components of metabolic pathways. These studies suggest this mouse model and interactome data will be useful to understand PC1 function, but that new methods and brighter tags will be required to track endogenous PC1.

## Introduction

The gene most commonly mutated in autosomal dominant polycystic kidney disease, *PKD1*, encodes a putative membrane protein with an ~3000 amino acid N-terminus, 11 transmembrane spanning (TM) domains, and an ~200 amino acid C-terminus [1]. This enigmatic but essential protein, PC1, undergoes a set of incompletely characterized post-transcriptional cleavage steps. The best understood occurs at its G-protein coupled receptor cleavage site (GPS) at the boundary of the 3000aa N-terminus and the first TM [2], resulting in an N-Terminal Fragment (NTF) and C-terminal Fragment (CTF) that remain tethered. Cleavage is thought essential for proper trafficking to the primary cilium [3] and homozygous mutations disrupting it result in severe, distal cystic disease and death by around P30 in mice [4].

**Funding:** This research was supported by the NIH, National Institute of Diabetes and Digestive and Kidney Diseases (NIDDK) Intramural Research Program, grant 1ZIADK075042. The funders had no role in study design, data collection and analysis, decision to publish, or preparation of the manuscript.

**Competing interests:** The authors have declared that no competing interests exist.

PC1 has been localized to multiple subcellular locations, including apical and basolateral membranes, endoplasmic reticulum [5–9], and its cytoplasmic C-terminus to the nucleus and mitochondria [10–12]. Likewise, multiple functions are reported for PC1, including flow-sensor [13], Wnt-receptor [14], atypical G-protein coupled receptor [15], and roles in cell-cell and cell-matrix signaling [16, 17], transcription regulation [10, 11] and mitochondrial function [12, 18].

Given PC1's large size and multiple domains, it is certainly possible that it localizes and functions in multiple compartments and pathways, but studies have reported conflicting results [12]. The discrepancies highlight one of the challenges in the field: tracking PC1 unambiguously. PC1 expression decreases dramatically in post-natal kidney tissue, and while its expression is higher in other adult tissues, less work has been done using other organs [19]. Numerous studies, therefore, have relied on over-expression of partial or full-length recombinant PC1, often with various epitope tags and unknown similarity to endogenous conditions. Two mouse models have been reported with the HA-epitope introduced into the endogenous mouse *Pkd1* locus [20] or into a Bacterial Artificial Chromosome (BAC) containing the mouse gene and introduced into the mouse as a transgene [21]. These *in vivo* systems provided direct evidence that the epitope tags had not seriously altered PC1 function, as the mice did not develop cysts. However, they proved inefficient for affinity-purification mass spectrometry (AP-MS), as studies failed to detect polycystin-2 (PC2) [16], the main PC1 interactor [22]. In addition, the HA tag does not allow live cell imaging of PC1 trafficking. Likewise, while our group had detected the C-terminus of PC1 (CTT) in mitochondria using cell-based imaging and biochemical methods, we were unable to unambiguously visualize endogenous PC1-CTT in mitochondria.

Given the conflicting results for PC1 and the general inability of the PKD community to visualize endogenous PC1, we sought to generate a new mouse line with an enhanced green fluorescent protein (eGFP) "knocked-into" the C-terminus of PC1 that could be used for visualizing endogenous PC1. In addition, we hypothesized that the availability of higher-affinity nanobodies to immunoprecipitated eGFP would allow for optimization of AP-MS protocols to identify PC1's endogenous interactome [23]. We also added a triple HA-epitope tag to allow dual purification/tracking of the endogenous protein using antibody-based detection systems. We report here characterization of the mouse and the first *in vivo* PC1 interactome.

## Methods

### Ethics statement

All mouse studies were performed using protocol ASP-K001-KDB-19, approved by the Animal Research Advisory Committee (ARAC) at the NIH Intramural Research Program. Mice were kept and cared in pathogen-free animal facilities accredited by the American Association for the Accreditation of Laboratory Animal Care and meet federal (NIH) guidelines for the humane and appropriate care of laboratory animals.

### *Pkd1* knock-in model

*Pkd1-GFP* knock-in mice were generated using the CRISPR/Cas9 technology. Briefly, one sgRNA (GGTCCACCCCAGCAGCACTT) targeting the last exon of *Pkd1* was designed to cut near the stop codon. The donor DNA construct was made by flanking the eGFP coding region, followed by 3-HA tags, with 5' (2992 bp) and 3' (3037bp) homologous arms of the cut site, introducing a silent mutation in the last threonine codon from ACT to ACA (to prevent the donor plasmid from being targeted by Cas9) and then inserting the GFP-3HA between the last codon and the stop codon (TAG). The donor DNA construct (10ng/ul) was co-microinjected

with Cas9 mRNA (50ng/ul) and sgRNA (20ng/ug) into the pronuclei of fertilized eggs collected from B6D2F1/J mice (JAX #100006). The injected embryos were cultured overnight in M16 medium overnight, and those embryos which reached 2-cell stage of development were implanted into the oviducts of pseudo-pregnant foster mothers (CD-1 mice from Charles River Laboratory). Offspring born to the foster mothers were genotyped by PCR. Two male mice with confirmed insertion were bred: one was a chimeric mouse with likely no germline insertion, and the other produced pups carrying the knock-in and was therefore our founder mouse. Genomic DNA of the founder mouse was used for long-range PCR of regions flanking the homology arms, GFP-3HA sequence, and all exons in the region, which were sequenced and confirmed to be without mutations (file with sequence and chromatograms available upon request). This animal was bred with C57BL/6 mice to establish the knock-in mouse line. F1 mice were either back-crossed to C57BL/6 mice (up to F4) or interbred. Subsequent genotyping of this line was done with the primers: E46-F: 5′-TGCTTGTCCAGTTTGACCGA with eGFP-R: 5′-GCTGAACTTGTGGCCGTTTA and 3UTR-R: 5′-ATGGCCACCTAGGGGTAGAG (wild type product: 614 bp; knock-in product: 332 bp). Both C57BL6J and C57BL6NJ were inadvertently used interchangeably through the initial phases of this work. C57BL/6J (JAX strain #000664) and C57BL/6NJ (JAX strain #005304) strains carry nicotinamide nucleotide transhydrogenase (*Nnt*) mutant and wild type alleles, respectively. *Nnt* genotype was determined by PCR as described in [24] and except when specifically noted, only *Nnt* wild type mice were used. Briefly, the Nnt PCR reaction identifies a 312bp product only in *Nnt* wild type mice using primers: Exon6_L1: CAATTCTGCCAACAACTGGA and Exon6_R4: GGTCACTCTGGGCACTGTTT; and a 547bp product only in mutants using Exon12_L1: GTAGGGCCAACTGTTTCTGC and Exon6_L4: TCCCCTCCCTTCCATTTAGT.

## Mouse studies

*Pkd1^{ko}* mice were previously described [25]. Mice homozygous for the *Pkd1^{ko}* allele lack exons 2 to 4 and die *in utero*. *Pkd1^{cko}* mice have loxP inserted into introns 1 and 4, and generates the deleted allele *Pkd1^{del2-4}* upon cre-expression [25]. Tamoxifen-inducible Cre-ER transgenic mice (JAX strain #004682 [26]) and Ksp-cre transgenic mice (JAX strain #012237 [27]) expressing cre recombinase in kidney tubular epithelial cells were used to induce *Pkd1* inactivation in the kidney. In the Cre-ER line, mice were induced at post-natal day 40 (P40) with 0.4mg/g tamoxifen (Sigma, T5648) diluted 20mg/ml in corn oil and injected intra-peritoneally under brief anesthesia with 3–5% isoflurane administered within a chamber. At the end of the studies, mice were euthanized with $CO_2$ administration in a chamber, followed by cervical dislocation. Throughout the study, mice were monitored and those showing signs of pain/suffering (such as hunched body posture and decreased food or water intake) were treated in consultation with the veterinary staff or euthanized as above.

## Primary cell lines

Primary tubular epithelial cells (PTEC) were obtained from kidneys removed from P2 pups. Briefly, kidneys were quickly dissected after mouse euthanization and transferred to gentle-MACS C Tubes (Miltenyi Biotec Inc., cat. no. # 130-093-237) in 2ml of digestion buffer (13.5 ml of Dispase (STEMCELL, cat. # 07912) with 1.5 ml of Collagenase/Hyaluronidase 10X (STEMCELL cat. # 0793), 150 μl of GlutaMax, 150 μl of 1M HEPES and 0.75 μl of 10 mg/ml DNAse I) and dissociated in gentleMACS Dissociator (Miltenyi Biotec Inc., cat. no. 130-093-235) using the Multi_E_01.01 program, followed by incubation at 37˚C for 30 min. under gentle rotation. Dissociated cells were filtered through pre-separation filters (Miltenyi Biotec Inc. cat. no. # 130-095-823) and centrifuged at 150xg for 5 min. and either directly seeded on 10cm

cell culture dishes or further enriched with *Lotus Tetragonolobus Lectin* (LTL; Vector labs bio-tynilated beads, cat. no. B-1325-2) LTL or *Dolichos Biflorus Agglutinin* (DBA; Vector labs bioti-nylated beads, cat. no. B-1035-5) beads and CELLection Biotin Binder Kit (Invitrogen, cat. no. 11533D). Cells were grown at 37˚C with 5% $CO_2$ in low serum media DMEM/F12 media (Life cat. no. 21041–025) with 2% FBS (GEMINI Bio-Products cat. no. 100–106), 1 x Insulin-Trans-ferrin-Selenium (Thermo Fisher Scientific, cat. no. 41400–045), 5 µM dexamethasone (SIGMA, cat. no. D1756), 10 ng/ml EGF (SIGMA, cat. no. SRP3196), 1 nM 3,3′,5-Triiodo-L-thyronine (SIGMA, cat. no. T6397) and 10 mM HEPES (CORNING, cat. no. 25-060-CI).

Mouse embryonic fibroblasts (MEF) were obtained from E13.5 embryos which were chopped coarsely (10–20 times) using a sterile razor blade and digested in 0.25% Trypsin-EDTA for 10 min. in a cell culture incubator (37˚C; 5% $CO_2$) followed by dissociation in gen-tleMACS C Tubes (Miltenyi Biotec Inc., cat. no. #130-093-237) in a total of 8 ml DMEM media with 10% FBS using gentleMACS Dissociator Program A twice and seeded onto 15 cm tissue culture dishes and grown in DMEM with 10% FBS at 37˚C with 5% $CO_2$.

## Immunoblot and immunoprecipitation

Mouse tissue was dissociated using gentleMACS Dissociator (Miltenyi Biotec Inc., cat. no. 130-093-235) and gentleMACS M Tubes (Miltenyi Biotec Inc., cat. no. # 130-093-236) in lysis buffer (1% Triton X-100; 150 mM NaCl; 50 mM Tris HCl pH8.0, 0.25% sodium deoxycholate, $1 \times$ protease inhibitor with EDTA (Roche Complete, cat. no. 11836153001), 1 x phosphatase inhibitor (Roche PhosSTOP, cat. no. 4906845001), 1 µl/ml Nuclease (Pierce, cat. no. 88700)). Immunoprecipitation of tagged PC1 was done using anti-GFP nanobodies coupled to mag-netic agarose beads (GFP-trap, Chromotek cat. no. gtma-10) following the manufacturer's pro-tocol and Kohli et al. [28] with the following modifications. After lysates and GFP-trap were rotating end-over-end for 1 hr at 4˚C, the GFP-trap/proteins complexes were separated from the lysates by a magnetic field. Non-specific binding proteins were washed away very gently and quickly (within seconds) in the cold room, twice with 500 µl of wash buffer 1 (50mM Tris (pH7.5), 150mM NaCl, 5% Glycerol, 0.05% Triton-X 100, $1 \times$ protease inhibitor with EDTA, 1 x phosphatase inhibitor) and wash buffer 2(50mM Tris (pH7.5), 150mM NaCl). For mass spectrometry, the GFP-Trap products were eluted in 160 µl of pre-heated (60˚C) 1% sodium dodecanoate (DODEC buffer) [29], transferred to the pre-heated glass vials, then heated at 60˚C for 20 minutes. For immunoblotting, samples were denatured in 1:1 with 2 x SDS buffer with 1x reducing agent (Invitrogen, NP0004) for 10 min at 95˚C, loaded into 4–12% bis tris 1.5 mm gels (Invitrogen, NP0336BOX), run in MES buffer (Invitrogen, NP0002) at 200V for 45 min., transferred to nitrocellulose membrane (Invitrogen, IB301002) using iBlot dry blot-ting transfer system (Invitrogen) program 3 and blocked with Intercept Blocking Buffer (Licor, cat. no. 927–70001), incubated with antibodies and imaged using Li-Cor Odyssey Imaging System following the manufacturer's instructions and the following antibodies: anti-eGFP (Aves, cat. no. GFP-1020), anti-HA (MBL, cat. no. M180-3), anti-NNT (Thermo Fisher Scientific, cat. no. 13442-2-AP or Santa Cruz Biotechnology, cat. no. sc-390215), anti-Polycys-tin-2 (PC2-CT [30], PKD-RCC (https://www.pkd-rrc.org/) and anti-Polycystin-1 (7E12, Santa Cruz cat. no. sc-130554), anti-β -actin (CST, cat. no. 4967S) and secondary labeled antibodies (Licor, cat. no. 925–68072 or 926–32211).

## Mass spectrometry

For each mass spectrometry experiment, 8 heads of P1 newborn mice of each genotype ($Pkd1^{eGFP/eGFP}$ and $Pkd1^{wt/wt}$) were used as input for immunoprecipitation (approximately 80mg of protein crude lysate/genotype). Briefly, two heads were combined, dissociated and

lysed as described above. Protein concentration was measured using BCA assay (Pierce, cat. no. 23225) and 20mg were used for immunoprecipation (IP) with GFP-trap. The eluates of the four independent IPs for each genotype were then pooled and immediately processed for mass spectrometry. This procedure was repeated three times (using a total of 24 heads). A bottom-up proteomic analysis used a soap to assist in rendering and digestion of sample proteins [29, 31], post-digestion reductive demethylation [32, 33], neutral pH off line reversed phase separation using fraction concatenation [34] conventional High-Low microscale HPLC/mass spectrometry, and processed with MaxQuant [35]. Given the expectation that there should be enrichment of pulled-down material in the samples containing the bait, we used non-normalized ratios for the analyses [36]. The mass spectrometry proteomics data have been deposited to the ProteomeXchange Consortium via the PRIDE [37] partner repository with the dataset identifier PXD035250 and 10.6019/PXD035250.

## Interactome analyses

Enrichment ratios of protein abundance in PC1-eGFP vs. control samples were obtained in MaxQuant [35] and imported into R [38] for further analyses. Gene set enrichment analyses using fgsea [39] was performed on the ordered values of the $\log_2$ of the mean ratios (across replicates). Pathway enrichment analyses were performed on lists of proteins with ratios above 1 in all experiments and ratio above 2 in at least two experiments (considered part of the PC1 extended interactome) and proteins sometimes enriched in PC1-eGFP, sometimes in the control groups, with ratios above ½ and below 2 (these were considered background proteins). These lists of interactome and background proteins were analyzed in Cytoscape [40] using stringApp [41] using default conditions. Networks were visualized in Cytoscape using yFiles layouts. Graphs were plotted in R using the ggplot2 package [42]. For color-coding and subsetting mitochondrial and ciliary proteins, we used goseq [43] and the org.Mm.eg.db database to extract genes in all gene ontology cellular component categories containing the keyword mitochondria or cilia. A list of previously published Polycystin 1 (PC1) interactors was obtained from publications [14–16, 44–49] and by querying PC1 in the STRING [50] 11.5 online database (http://string-db.org/) as an output that represents the associations in the form of a physical subnetwork with nodes (proteins/genes) and edges (interactions) to extract the 1st order interactomes. The 1st order interactome contained a max number of 50 interactors. The edges are weighted and integrated, and a confidence score is assigned to each of them based upon the evidence of the association obtained from text mining, experimental data and public databases. The high (0.70)/medium (0.40)/low (0.15) confidence scores in the database, which define and compare the significance of interactions between various queried proteins, were chosen respectively to extract STRING-based PC1 interactomes.

## Histology and immunostaining

Cells were seeded at a density of 100,000 cells/well in chamber slides (Ibidi, cat. no. 80822). After 24h, cells were serum-starved for 48h, fixed for 10 min. in 10% formalin, rinsed with PBS and permeabilized with 0.5% Triton X-100 in PBS for 5 min., blocked with 1% fish gelatin in PBS, stained with antibodies to detect Arl13 (ciliary marker; ProteinTech cat. no. 17711-1-AP; secondary antibody: Invitrogen A27034 labeled with Alexa Fluor 488), γ-tubulin (ciliary basal body marker; SIGMA cat. no. T5192; secondary antibody Invitrogen A27034 labeled with Alexa Fluor 488) or eGFP (Aves cat. no. GFP-1020; secondary antibody: Invitrogen A21449 labeled with Alexa Fluor 647), mounted (Invitrogen ProLong cat. no. P36981) and imaged using Zeiss LSM700 confocal microscope. Freshly dissected embryos or kidneys were frozen in OCT (Tissue-Tek; Sakura Finetek USA) and frozen sections were processed as above; or

embedded in paraffin, sectioned, and stained with trichrome staining or used for immunohistochemistry. Images were visualized using Imaris (Bitplane). Cystic index was calculated by delineating kidneys in trichrome stained slides using the magnetic lasso feature in Photoshop, and measuring total kidney area, thresholding cystic area and measuring total kidney area and cystic area using Fiji [51].

### Statistics

The R environment [38] was used for statistical analysis. To minimize the effect of outliers, robust statistical estimators using the WRS2 package were used [52]. In particular, comparisons of two groups were done using the function pb2gen for t-test based on medians. Plots were made using the ggplot2 package [42]. The pedigree tree network was generated in Cytoscape [40].

## Results

### Generation and characterization of *Pkd1* knock-in model

Using CRISPR/Cas9 technology, we inserted eGFP followed by three human influenza hemagglutinin (HA) tags in frame with PC1 C-terminus (Fig 1A–1C). Screening of targeted mice revealed one mouse with germline knock-in carrying this expected $Pkd1^{eGFP}$ allele, confirmed by long-range PCR and RT-PCR (Fig 1D and 1E). This animal was the founder of the $Pkd1^{eGFP}$ knock-in line (Fig 1F). Immunoblot analyses showed that tagged polycystin-1 (PC1-eGFP) is expressed in kidneys of young pups, with detection using both the HA-tag and antibodies to the endogenous protein (Fig 1G). Using nanobodies against eGFP, PC1-eGFP can be efficiently enriched in pull-down experiments (Fig 1H–1J) and can be detected in multiple mouse tissues, including brain, lung, heart and kidney; and in primary cell lines obtained from knock-in mice.

The founder mouse was bred to wild type and to heterozygous knock-out ($Pkd1^{ko/wt}$) mice (Fig 2A). $Pkd1^{eGFP/wt}$ bred to $Pkd1^{ko/wt}$ produced pups at the expected mendelian rates (Fig 2B). A cohort of 6 $Pkd1^{eGFP/ko}$ mice was aged to 15 months and showed no kidney pathology, though one animal developed small liver cysts (Fig 2C–2E). $Pkd1^{eGFP/eGFP}$ mice euthanized at multiple ages between birth and 450 days of age had no obvious phenotypes (>10 mice/30-day brackets). These data suggest that a C-terminal tag added to PC1 does not significantly impair its function.

One of the goals of creating a *Pkd1* knock-in mouse was to detect PC1 *in vivo* using microscopy. So far, we have been unable to unambiguously identify PC1 in live cells/tissue or in fetal or adult tissues using antibodies that detect the HA or eGFP tags. However, we have consistently detected PC1 using antibodies to GFP in the primary cilium of kidney epithelial cells (Fig 3A and 3B) and some cilia in mouse embryonic fibroblasts (Fig 3C). We have been unable to detect PC1 unambiguously above background in other subcellular structures.

### *In vivo* PC1-interactome

Another goal of the *Pkd1* knock-in model was to identify *in vivo* PC1 binding partners and to establish a PC1-interactome. Towards this goal, we optimized conditions to immunoprecipitate PC1 and detect both PC1 and PC2, a known PC1 binding partner [53]. Having confirmed this in mouse tissue (S1 Fig), we tested multiple protocols and found that reliable detection of PC1 and binding partners by mass spectrometry in IP isolates required approximately 80mg of mouse tissue. We then performed 3 sets of affinity purification mass spectrometry experiments (Fig 4A), each analyzing immunoprecipitates from the lysate of a combination of eight P1

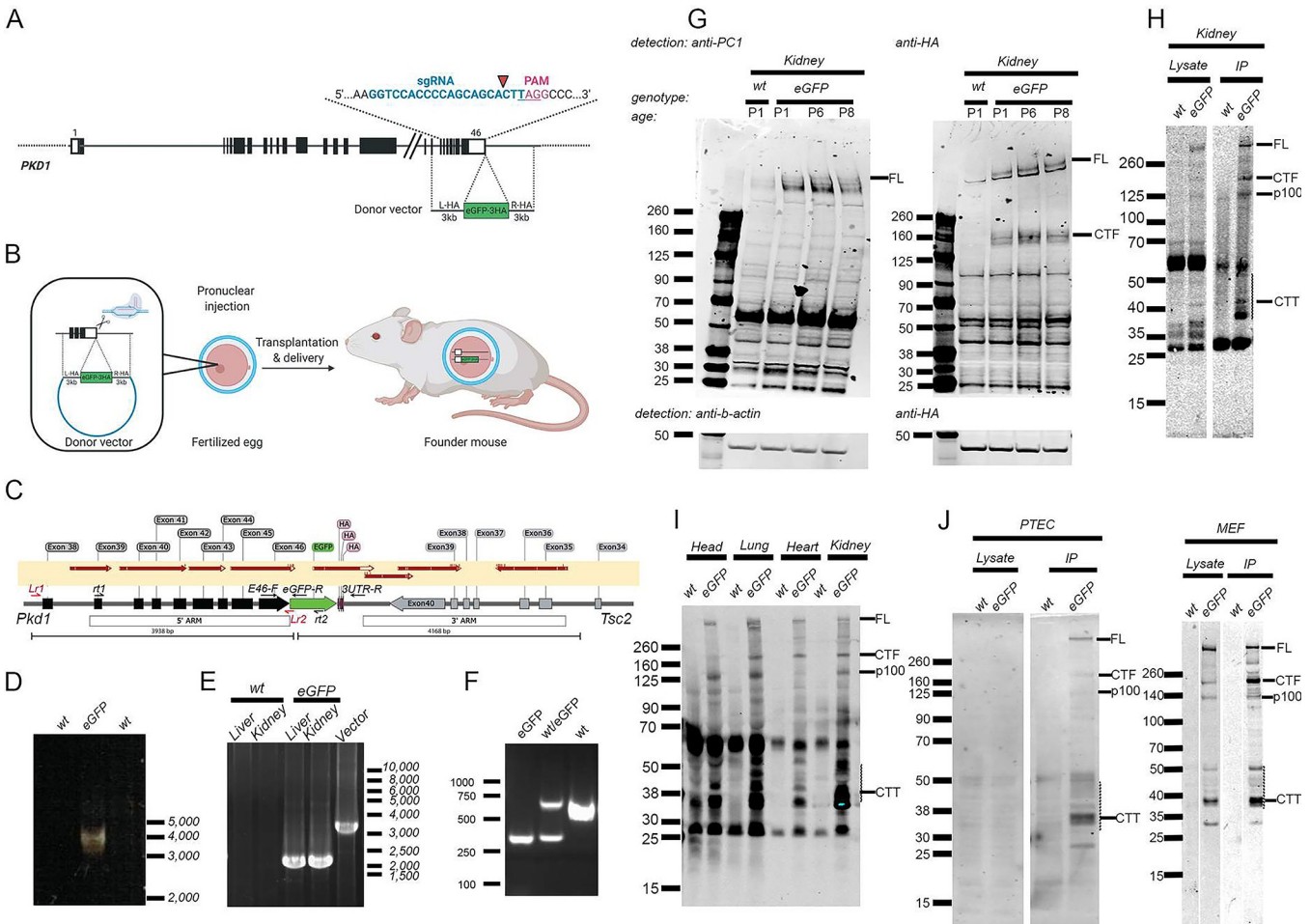

**Fig 1. *Pkd1^eGFP* knock-in mouse generation. A)** Schematics of *Pkd1* knock-in allele, showing sgRNA guide sequence and location of eGFP-3xHA tag. Created with BioRender.com. **B)** Diagram showing steps to generate the knock-in mouse. **C)** Structure of the knock-in *Pkd1* allele. Red arrows on the light yellow rectangle mark regions confirmed by sequencing. Small arrows in the gene diagram are the locations of primers used in the reactions in panels D-F. **D)** Genomic long-range PCR using primers Lr1 and the eGFP-specific Lr2 showing the predicted band of ~4kb in *Pkd1^eGFP* founder mouse. No bands are amplified in the control tissue (wt). **E)** Reverse-transcriptase PCR using primers rt1 and rt2 showing expression of the *Pkd1* transcript with eGFP sequence in knock-in mouse tissue. The vector positive control is larger because it includes introns. **F)** Genomic PCR using primers E46-F, eGFP-R and 3UTR-R to genotype knock-in mice. As expected, eGFP homozygotes only have the ~300bp E46-F:eGFP-R product, control specimens (wt) have only the 600bp E46-F:3UTR-R product, and heterozygotes have both. **G)** Detection of PC-1 in kidney of wild type and knock-in mice at different post-natal days; left: detected with anti-HA antibody; right: detected with anti-PC1 (7e12) antibody. Full length, uncleaved PC1 is detected with the anti-HA antibody (left) whereas 7e12 only detected NTF under these conditions (right). β−actin was used as a loading control; 7e12 specificity is shown in S1 Raw images). **H)** Kidney of control and knock-in P1 mice showing tagged PC1 detection with anti-HA antibody; left: cell lysate; right: protein immunoprecipated with anti-eGFP conjugated agarose beads. Note the enrichment of CTT after IP. PC1 full-length (PC1_FL), cleavage products (PC1_CT; P100), and multiple CTT-related cleavage products (presumed CTT marked with an arrow; dotted line represents range of possible CTT products). **I)** Immunoprecipitation of PC1 from different tissues of P1 control and knock-in mice using anti-eGFP conjugated agarose beads and detected with anti-HA antibodies. **J)** Mouse primary tubular epithelial cells (PTEC) and embryonic fibroblast (MEF) cells from control and knock-in mice showing tagged PC1 in the lysate (left) and immunoprecipitated by anti-eGFP and detected with anti-HA (right).

mice heads. We selected P1 mice head because brain had one of the highest PC1 expressions at this age in both knock-in and control mice. As expected, under these conditions, both PC1 and PC2 were the proteins with highest enrichment and intensity in knock-in mouse samples compared to controls (up to 36 times) (Fig 4B, S1 Table).

The protocol was optimized with low-stringency, fast washes, with the goal of immunoprecipitating (IP) weak interactors [54]. Consequently, both experimental (PC1-eGFP) and control IPs from wild type mice lacking the eGFP-HA tag identified similar proteins, reflecting a

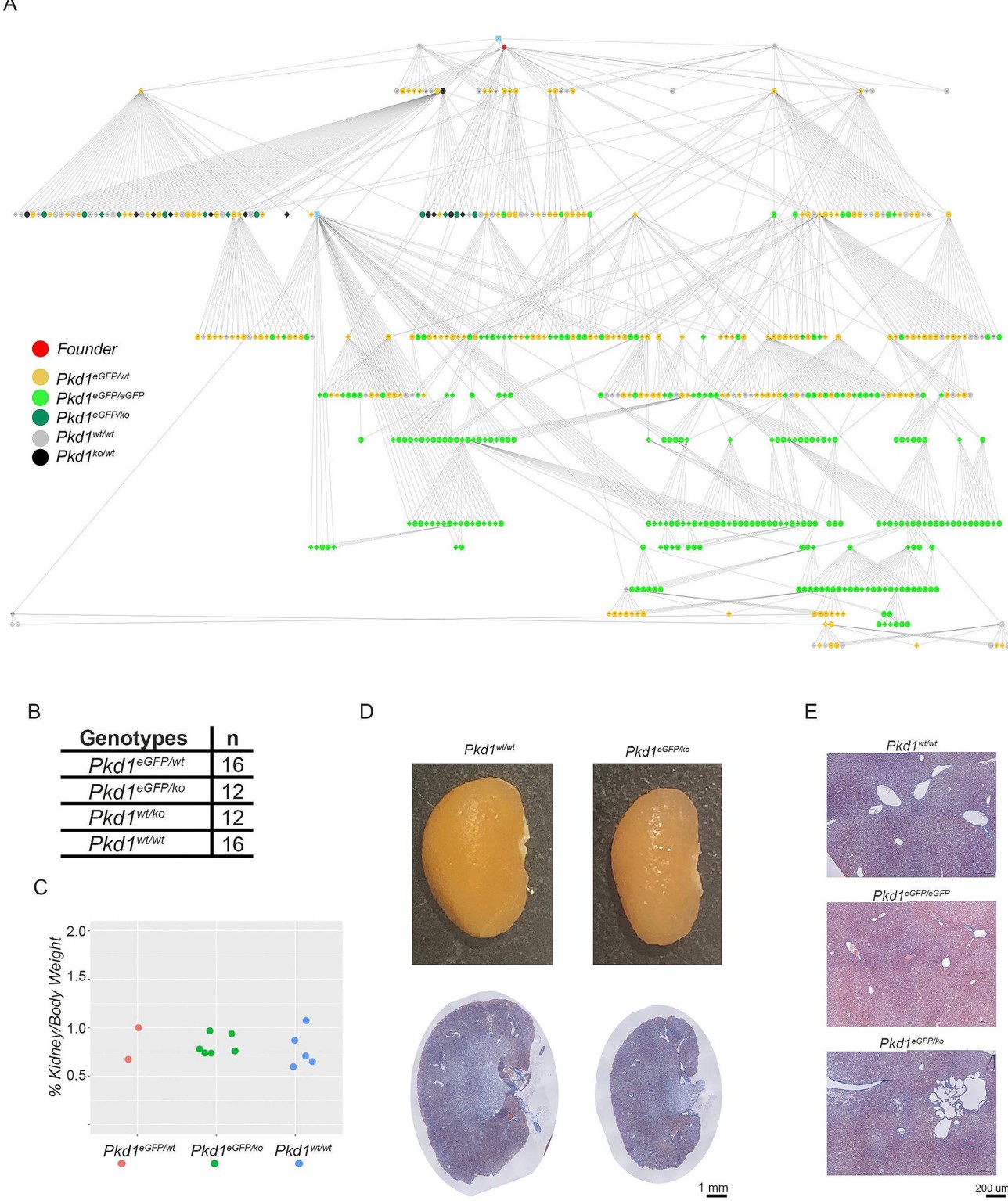

**Fig 2. *Pkd1^eGFP^* knock-in mouse. A)** Pedigree tree of mouse colony derived from the single *Pkd1^eGFP^* knock-in mouse (red dot at the top). Genotypes are color-coded. The blue dots represent wild type mice used for back-crossing. **B)** Table showing that *Pkd1^eGFP/ko^* pups born to *Pkd1^eGFP/wt^* vs. *Pkd1^ko/wt^* breedings are viable. **C)** Kidney/body weight ratios of a subset of animals in panel B allowed to age to 15 months showing normal kidney size. **D)** Representative *Pkd1^eGFP/ko^* kidney has normal morphology at 15 months of age (lower panels: trichrome staining; size bar: 1mm). **E)** While the majority of the *Pkd1^eGFP/eGFP^* and *Pkd1^eGFP/ko^* livers had normal morphology, one 15 months-old *Pkd1^eGFP/ko^* mouse had liver cysts (middle panel; trichrome staining; size bar: 200 μm).

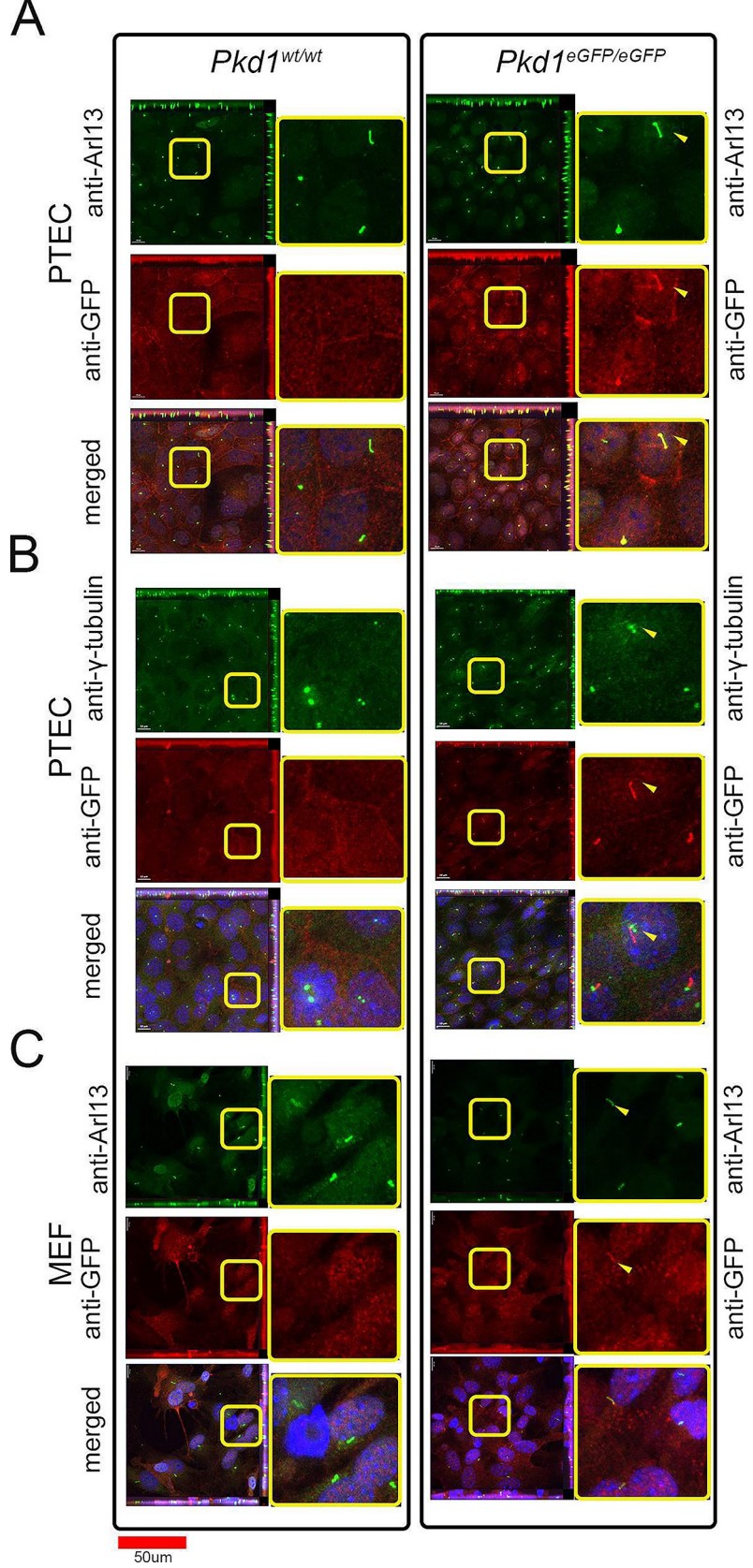

**Fig 3. Tagged PC1 can be detected in cilia of primary cells and tissues of knock-in mice. A)** Detection of PC1 with anti-eGFP antibody in cilia of most primary tubular epithelial cells (PTEC), co-localized with Arl13; **B)** PC1 localizes adjacent to ciliary basal bodies, marked with anti-γ-tubulin. **C)** PC1 is detected in cilia of some, but not all, mouse embryonic fibroblasts (MEF) cells.

somewhat high background (S1 Table). A standard way to identify true interactors under these conditions is to focus on consistent enrichment across replicates in experimental vs. control conditions [54]. Using this method, we identified 83 proteins with a ratio >2 in all three experiments (Fig 4B). For this study, however, we considered part of the PC1 extended interactome all proteins that were enriched (ratio > 1) in all PC1-eGFP IP mass spectrometry experiments (n = 3) and had a ratio of at least 2 in at least 2 of those (S1 Table). With these criteria, 823 proteins were identified (S2 Table).

## PC1-interactome is enriched in metabolic pathways and mitochondrial proteins

Using stringApp [41] in Cytoscape [40], we obtained the protein network of reported physical interactions within these proteins in the STRING database [50] and identified a large cluster consisting of 522 proteins linked by over one thousand interactions, consistent with our expectation that the immunoprecipitated proteins were part of an extended network of interactors (Fig 5A). Pathway analyses showed enrichment of cytosolic ($p<10^{-48}$) and mitochondrial proteins ($p<10^{-25}$) (S3 Table). Using a set of likely background proteins (the 991 proteins with no ratio above 2 or below ½ that were sometimes enriched in PC1-eGFP and sometimes in control samples), similar analyses showed that cytosolic and mitochondrial proteins were among those with highest change in significance between the background and interactome sets (Fig 5B), whereas nuclear proteins were enriched in the background, but depleted in the interactome. While ciliary and cytoskeletal proteins were detected in the interactome (S2 Fig; S2 Table), neither category was enriched. Functionally, the PC1 interactome was enriched in oxidation-reduction pathways ($p<10^{-18}$) and several of the metabolic pathways described in Wikipathways [55] (Fig 5C and 5D), including proteins involved in TCA cycle, glycolysis and fatty acid metabolism (Fig 5E), consistent with PC1 being a part of a network of proteins involved in metabolic and mitochondrial functions. Among signaling cascades, MAPK signaling was significantly enriched (Fig 5E). Gene set enrichment analysis (GSEA) of the interactome identified NADP metabolism as one of the top pathways (Fig 5F) and NAD(P) binding domains (Fig 5G) as the top enriched InterPro domain [56].

Previous reports have suggested mitochondrial dysfunction and reactive oxygen species (ROS) damage in PKD [18, 57–60]. GSH is a scavenging antioxidant that forms one of the main lines of defense against reactive species in most cells and NADPH supply is reported to be limiting in GSH synthesis [61, 62] in certain conditions. One of the proteins in the interactome involved in NADP metabolism is nicotinamide nucleotide transhydrogenase (NNT), an inner mitochondrial membrane protein that uses $H^+$ re-entry into the mitochondrial matrix and NADH to reduce $NADP^+$ into NADPH. In fact, among the 161 proteins in the PC1 extended interactome that were in mitochondrial gene ontology categories (S2 Table; Fig 6A), NNT was one of the top hits. It was enriched in all replicates, represented by peptides spanning the length of the protein (S3B Fig) and confirmed in PC1-immunoprecipitates of P1 heads and P2 kidneys of the PC1-eGFP knock-in mice (Fig 6B).

There are two commonly used C57BL/6 mouse lines that differ with respect to their Nnt status. The C57BL/6NJ line carries wild type *Nnt* alleles while the C57BL/6J lines has an exon 1 missense mutation and a multiexon deletion that severely compromises Nnt function [24, 63].

A

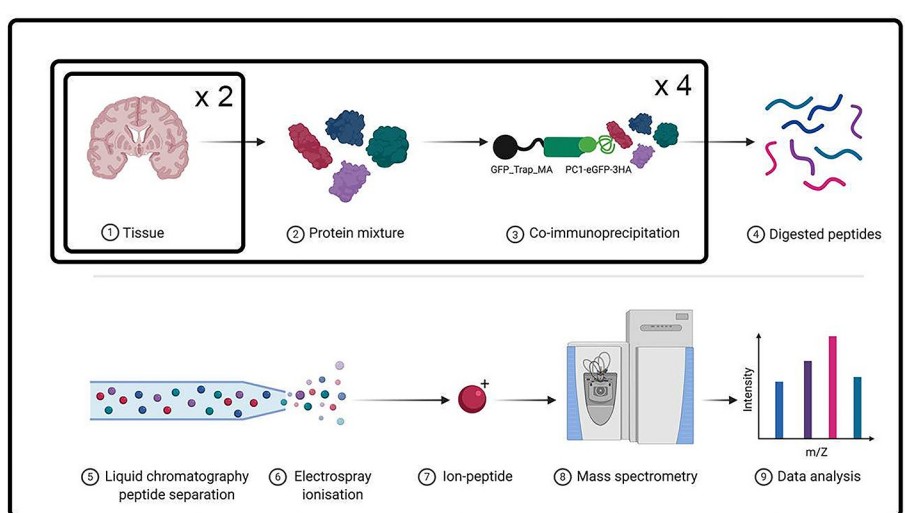

B

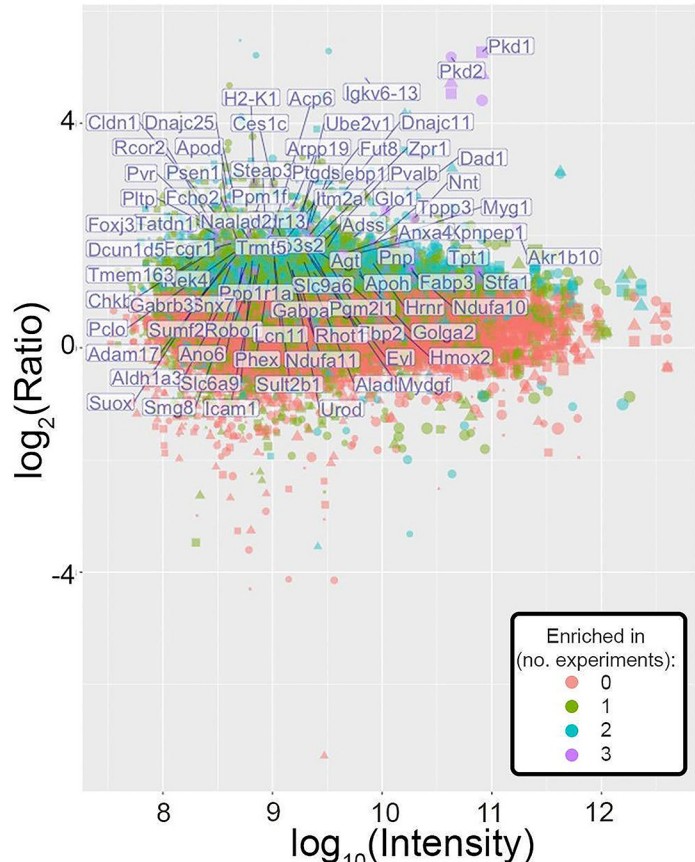

**Fig 4. Identification of PC1 *in vivo* binding partners. A)** Schematics showing steps for proteomics experiments. Briefly, for knock-in and control samples, 4 sets of 2 heads of P1 mice per genotype were independently immunoprecipitated using eGFP beads then pooled and processed for one mass spectrometry experiment. Each mass spectrometry experiment was done three independent times. Modified from template created with BioRender.com. **B)** Plot showing the $\log_{10}$(Intensity) (x-axis) and $\log_2$(intensity ratio between knock-in and control samples) (y-axis) prepared from P1 mouse heads. Each dot corresponds to one detected protein; shape (circle, square, triangle) represents experimental batch; the colors summarize the number of times the enrichment ratio was above 2 for the protein. Proteins that were never enriched in the knock-in IP are orange; proteins enriched in all three experiments are colored purple and are identified by purple text. PC1 and its primary binding partner PC2 were reproducibly the top hits in each IP study.

We had the *Pkd1*$^{cond}$ allele in both the C57BL/6J and C57BL/6NJ strains in our colony so we investigated if a PC1-NNT interaction could have functional consequences by comparing kidney disease severity in an early-onset (Ksp-Cre) and late-onset (inducible tamoxifen-cre) disease model. In the Ksp-cre model, at age 8 days (P8) *Pkd1* mutants have highly cystic kidneys but there was no significant difference between the *Nnt* wild type and mutant strains, though disease severity was more variable in the *Nnt* wild type background (Fig 7A and S4 Table; n = 10 *Nnt* mutants and 34 controls). We also saw no effect of Nnt status on disease severity in mice induced at P40 and harvested between P180 and P200 (Fig 7B and S5 Table; n = 6–7 per group).

These data suggest that interaction between PC1 and Nnt, also recently reported by another group [64], is not *per se* sufficient to significantly alter the cystic phenotype in *Pkd1*$^{-/-}$ between *Nnt* wild type and mutant strains.

### Previously reported PC1 interactors in the PC1 interactome

We also compared the extended PC1 interactome with a list of proteins previously reported to be binding partners of PC1. We used two approaches to identify targets: a) the STRING database, selecting targets either experimentally validated, present in curated databases or identified through text mining; b) targets reported in the literature but not necessarily present in STRING (S6 Table). We identified a total of 7 proteins present in both our dataset and other sources. Two were in the mouse proteome, 1 in human, and three in both mouse and human (Fig 8). The seventh was solely in *Xenopus*. Except for PC2, none of the hits was in the set of 84 proteins enriched >2 in all experiments.

Given that the extended PC1-interactome was greatly enriched for mitochondrial and cytoplasmic targets, we reviewed the pattern of PC1 peptides isolated by our IP method to exclude a CTT bias. As expected, given that full-length PC1 was detected by immunoblot in the IP eluent (S1 Fig), peptides in our mass spectrometry studies mapped along the length of PC1 (S3 Fig). These data suggest that our strategy should have captured targets for any of the domains of PC1, excluding a bias towards enrichment of just CTT targets.

### Discussion

PC1 has been slow to give up its secrets since the description of its primary sequence in 1995. Both its function and subcellular localization remain incompletely defined with conflicting reports adding to the uncertainty. We reasoned that unambiguous localization of PC1 and identification of its interactome would provide a sound basis for assessing its function.

We therefore used CRISPR/Cas9 [65] to add, in-frame to the end of the C-terminus of PC1, three HA epitope tags and eGFP, a particularly attractive fluorescent tag given its brightness and the availability and development of nanobodies for its affinity purification with very high specificity and affinity [66, 67]. We confirmed correct integration and expression of full-length PC1 with HA and eGFP tags. Mice either homozygous for the modified *Pkd1* allele or having it

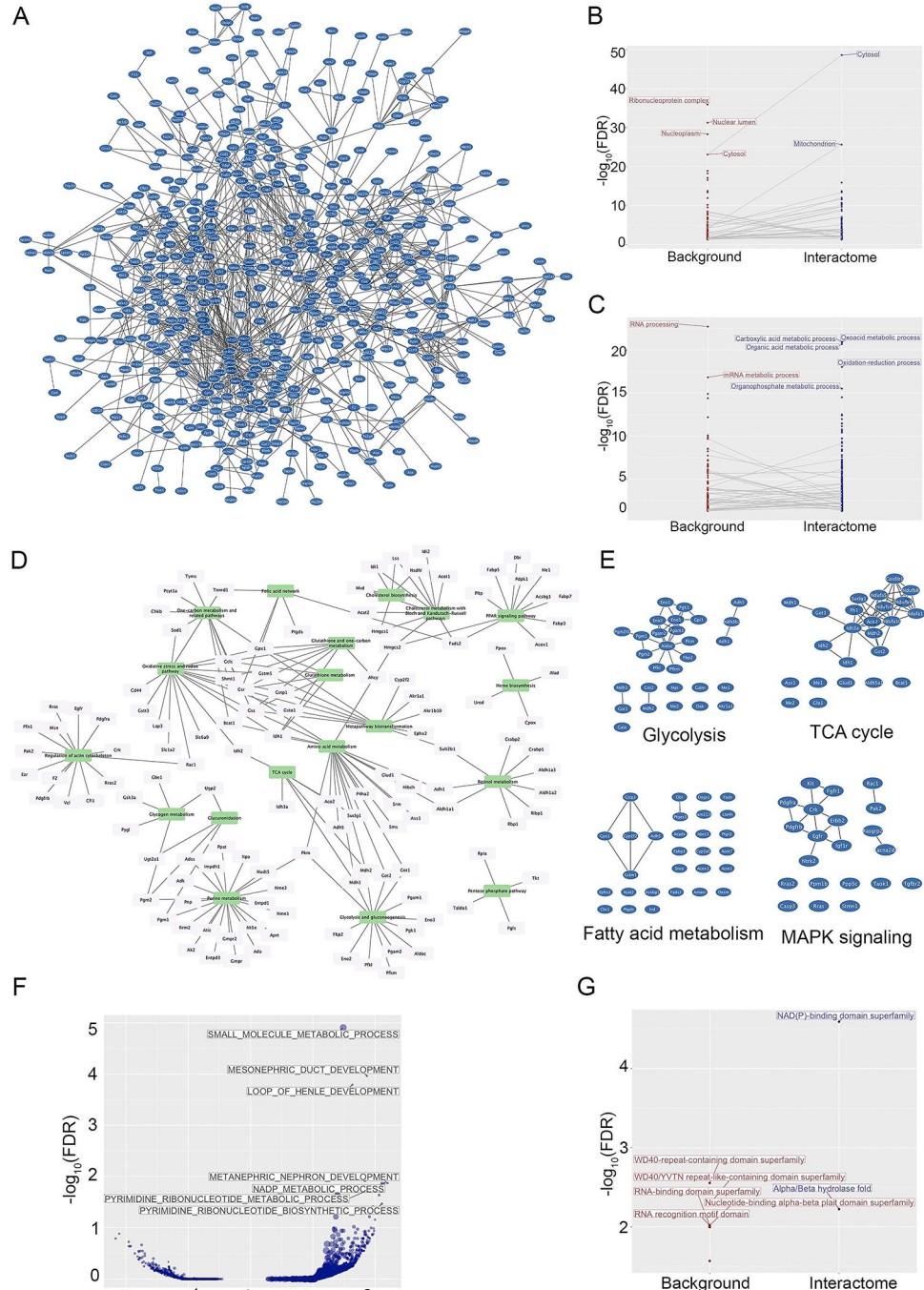

**Fig 5. PC1 interactome. A)** Physical interactions described in the STRING database connecting a large cluster (522 nodes) of the interactome. **B and C)** Pathway analyses of the extended PC1 interactome compared to background proteins using the Gene Ontology Cellular Component (D) and Gene Ontology Biological Processes (E). **D)** Network showing Wikipathways (green nodes) significantly enriched (p<0.05) in the interactome and the corresponding proteins (clear nodes). **E)** Physical interactions described in the STRING database connecting proteins in the PC1 extended interactome that were in some of the enriched pathways. **F)** Plot of Gene Ontology Biological Processes categories identified in gene set enrichment analysis (GSEA) of the extended PC1 interactome. Each dot corresponds to a category and its size correlates with the number of proteins in the category. The axes show -log$_{10}$(FDR-adjusted p-value) and normalized enrichment score (positive for categories enriched in the interactome). Note that NADP metabolic processes is one of the top enriched categories. **G)** Pathway analyses of the extended PC1 interactome compared to background proteins using the InterPro domains database. Note the enrichment of NAD(P) binding domains.

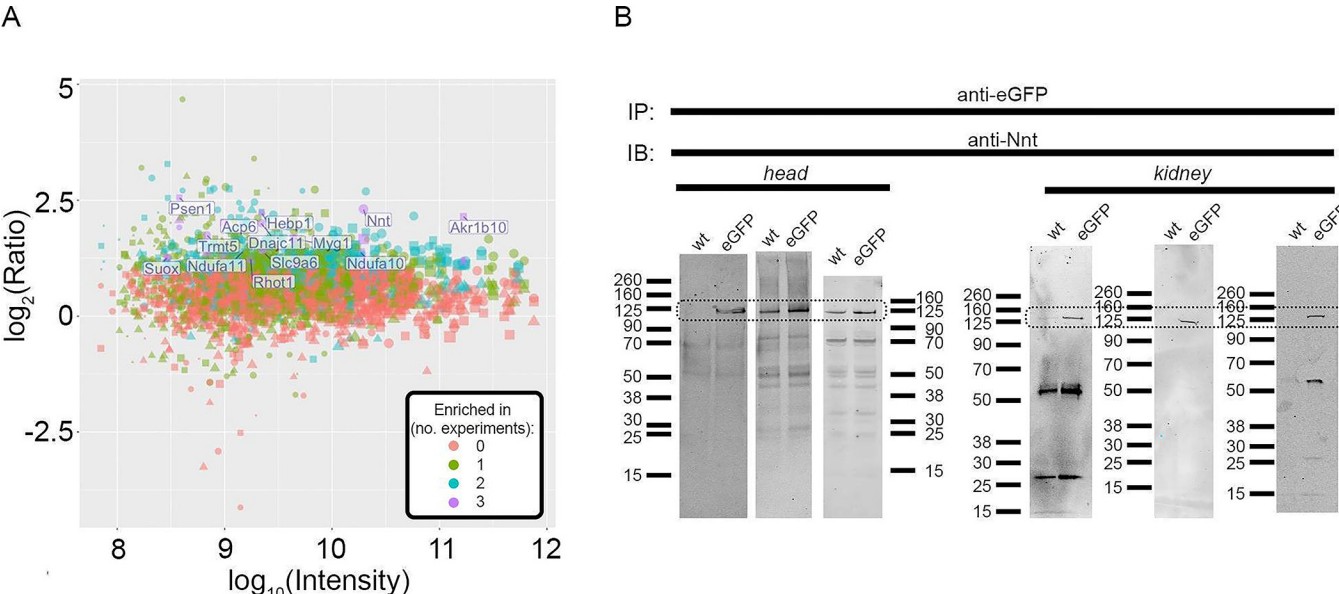

**Fig 6. NNT is likely a PC1 interactor. A)** Plot of immunoprecipitated proteins belonging to mitochondrial Gene Ontology Cellular Component categories. The axes show the $\log_{10}$(intensity) and $\log_2$(intensity ratio between knock-in and control samples). Each dot corresponds to one detected protein; shape (circle, square, triangle) represents experimental batch; the colors summarize the number of times the enrichment ratio was above 2 for the protein. Proteins that were never enriched in the knock-in IP are orange; proteins enriched in all three experiments are colored purple. Note NNT is one of the top reproducibly immunoprecipitated mitochondrial proteins. **B)** Immunoblot showing protein immunoprecipitated using anti-eGFP nanobody in control (wt) and *Pkd1*[eGFP/eGFP] P1 heads and P2 kidneys and detected with anti-NNT antibodies. The doted box indicates the region with NNT bands. Note that immunoprecipitation conditions for head and left kidney panels followed low-stringency washing conditions used for the mass spectrometry studies, resulting in variable levels of NNT detection in control samples, albeit with overall enrichment in *Pkd1*[eGFP/eGFP] samples compared to control samples. The washing and blocking conditions for the P2 kidney sample on the right panels were more stringent.

"in trans" with a *Pkd1* null allele were born at expected mendelian frequencies and aged normally without developing renal cystic disease. These findings definitively show that the addition of this relatively long "foreign" protein sequence had minimal effect on PC1 processing or function. It is worth noting that this is unlikely explained by rapid cleavage and release of eGFP-3HA from the rest of PC1, as the sequence is present in both full length PC1 and its various normal cleavage products (CTF, P100, CTT). We did observe one aged *Pkd1*[eGFP/ko] mouse

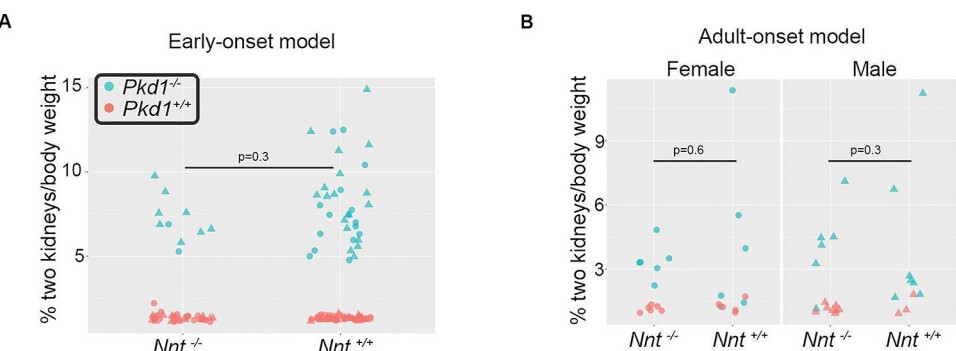

**Fig 7. *Nnt* status and PKD progression. A and B)** *Nnt* status does not correlate with kidney/body weight. Kidney/body weight in *Pkd1* mutants (blue) and controls (red) by *Nnt* status in P8 mice induced with nephron-specific Ksp-cre (B; n = 10 *Pkd1/Nnt* mutants and 34 *Pkd1* mutant/Nnt controls) or induced with tamoxifen at P40 and harvested between P180-P200 (n = 6–7 of each sex/group). Sex of each mouse is identified by its shape (triangles: males; circles: females).

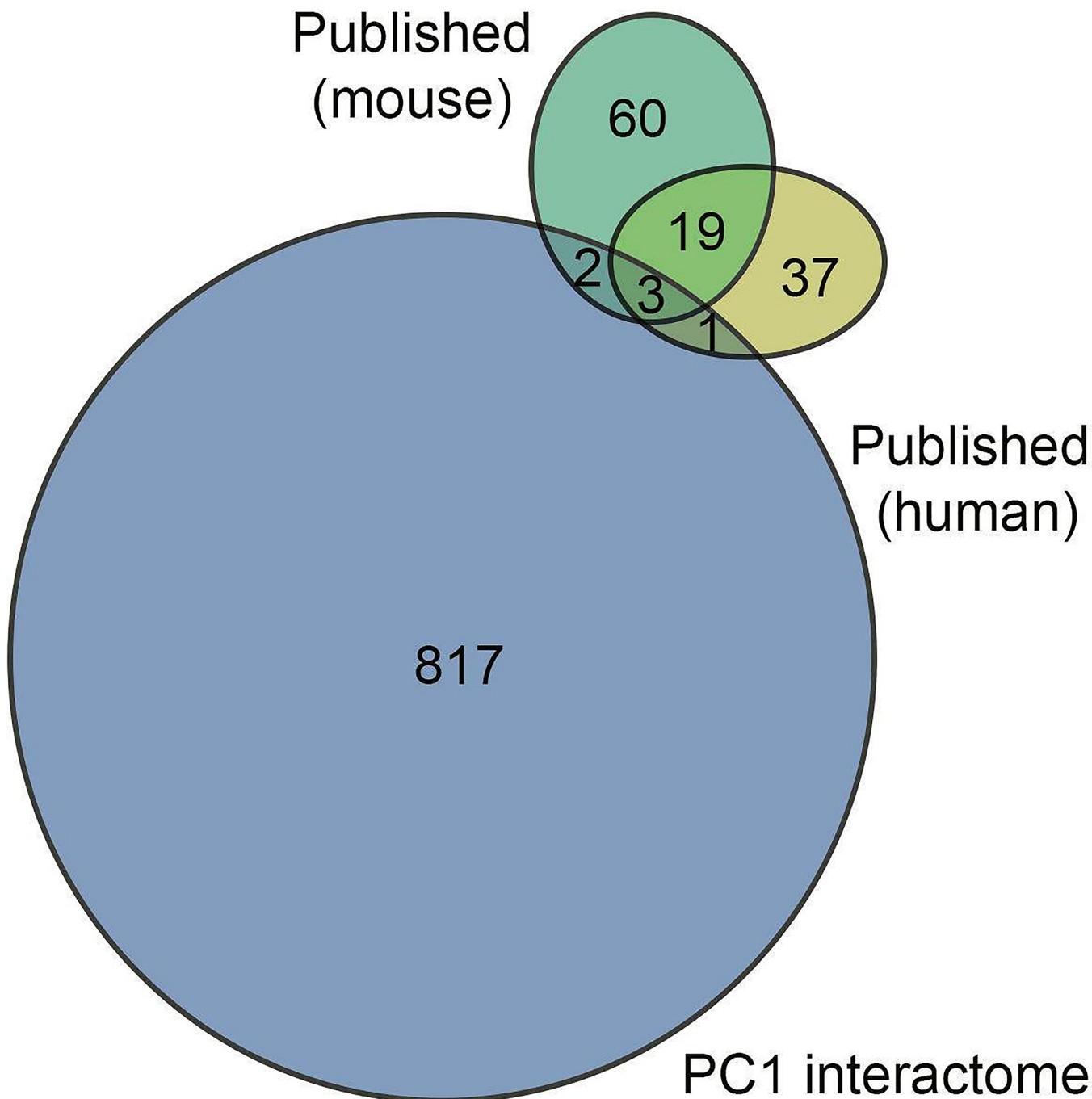

**Fig 8. Overlap of PC1-interactome with previously identified PC1-interactors.** Venn diagram showing the overlap between the PC1-interactome with targets previously reported for mouse and human (one interactor identified in *Xenopus* is not shown).

with mild cystic liver disease, suggesting that under certain conditions the added sequence might slightly impair PC1 function, but we conclude that in general the new allele functions properly.

One major disappointment was our inability to track PC1 in tissues or cells of mice homozygous for the modified allele without antibody amplification, suggesting PC1 expression is

very low and/or dispersed broadly so unable to detect above background because of lack of distinct clustering. While unambiguous tracking of endogenous PC1 seems likely to remain challenging for the foreseeable future, our studies suggest PC1 can tolerate the addition of a sizeable polypeptide to its C-terminus so perhaps alternative labeling methods using enzymes like the HaloTag or SNAP-tag may also be well-tolerated, though it is unknown whether these would improve signal-to-noise ratio.

Knowing what proteins are associated with one's target can provide useful clues into its functions [68]. We designed the new *Pkd1* allele with the goal of being able to characterize the endogenous interactome of PC1. We worked through conditions to determine the amount of tissue necessary to ensure that PC1 and its known interactor, Polycystin-2, were consistently among the top hits, a necessary minimal standard. This required a substantial scale-up of starting material, far beyond what is typically used. This observation further illustrates just how low endogenous PC1 expression is in tissue and why its characterization has been so challenging.

For the present study, we used P1 mouse head as the starting material because it is one of the largest structures in young mice and has high PC1 expression, but even this approach required eight mice for a single IP. While kidneys would have been preferable, this would have required many more mice. We reasoned that protein targets of interest could be later validated in kidneys, an assumption subsequently confirmed.

We intentionally used relatively low stringency washes to maximize the likelihood of capturing weak interactions. Our approach succeeded in reproducibly identifying several potential targets, many in metabolic pathways and over 60% of which physically connected in a single cluster, suggesting that most of the interactome identified in this study is comprised of weak or transient direct interactors or pathway components bound to direct interactors.

One surprising result was that PC2, the only protein definitively shown to form a complex with PC1, was also the only target reproducibly identified with very high enrichment compared to the control. There was a small number of other proteins enriched up to >10 times in one or two IP studies, but it is unknown whether these are direct interactors that form less stable complexes with PC1; bind indirectly to PC1 through an intermediate; or are just "false positives".

Another surprise is the small number of previously reported PC1 binding partners found in this dataset. We verified that our approach captured full length PC1, ruling out a C-terminal bias introduced by using antibodies targeting the C-terminus as a possible explanation. Perhaps this discrepancy reflects differences in methods, tissue source, or using endogenous rather than over-expression of recombinant PC1 as bait, but the relatively low stringency of washing should have maximized the likelihood of retaining true targets. Studies of other organs may help resolve this issue.

Full characterization of PC1 interactors is beyond the scope of the present work, but we did investigate network properties of the interactome. Our analyses show that several proteins in this PC1 interactome also interacted with each other. Since our immunoprecipitation protocol was aimed at increasing sensitivity rather than specificity, it is possible that PC1 is a direct binding partner of only a subset of the proteins in this interactome; the remaining being indirect partners or part of complexes that interact with PC1. In this scenario, the enrichment of multiple metabolic and mitochondrial proteins is telling. It suggests that PC1 could be a hub linking several physiological processes in the mitochondrion and is consistent with our previous report that PC1-CTT is a mitochondrial protein.

One aspect of this hypothesis we indirectly investigated in this report was NADP metabolism. This was in part because GSEA of the interactome had identified as one of the top pathways and NAD(P) binding domains as the top enriched InterPro domain. NADPH/NADP

+ play roles in ROS damage and signaling that have been linked to PKD [12,18, 57, 58]. *Nnt*, an important player in the pathway, was among the top hits in our mitochondrial proteins and was immunoprecipitated with PC1 in kidney samples. Investigating if Nnt could be a disease modifier, our results showed that Nnt status by itself had no effect on kidney cyst development. Why we saw no effect is unclear, but multiple explanations are possible: 1) the effect may be modest and obscured by the large variability in disease presentation; 2) mice born lacking *Nnt* may have compensation by other factors; 3) NNT's function in the kidney depends both on the presence of PC1 and other pathways regulated by PC1: loss of NNT in the presence of PC1 would be inadequate to trigger cyst formation whereas loss of NNT in the absence of PC1 would make no difference; 4) additional perturbations in the pathway may be required to uncover an Nnt effect, as suggested by recent reports of PC1-CTT rescue of PKD phenotypes only in *Nnt* wild-type mice [64]; 5) finally, we cannot exclude that NNT's interaction is functionally irrelevant.

In sum, we have produced a new model of PC1 with a fluorescent tag knocked in-frame into the C-terminus of PC1 and used it to show that PC1 expression is too low to detect reliably above background using microscopy, and efforts to track endogenous PC1 expression will require more sensitive methods. This new mouse model can be used, however, to study the PC1 interactome if sufficiently large amounts of starting material are used. We suggest that as a minimal standard all future interactome studies based on IP methods should detect PC1 and PC2 among top targets if PC1 is being used as bait. We have generated the first interactome of endogenous PC1 from mouse tissues and determined that it is enriched for components of metabolic pathways and mitochondrial targets. Finally, we have identified NNT as a probable bona fide PC1 interactor, further supporting a role for PC1 in the regulation of mitochondrial function.

## Supporting information

**S1 Fig. PC1 immunoprecipitation from post-natal day 1 (P1) whole mouse body.** The left panel PC1 shows full-length (PC1_FL), cleavage products (PC1_CT; P100), and multiple CTT-related cleavage products (presumed CTT marked with an arrow; dotted line represents range of possible CTT products). The right panel is the same blot, probed with anti-PC2 and detected in a different wavelength, showing PC2 co-immunoprecipitation. PC2 appears to occasionally form oligomers depending on denaturing conditions (unpublished previous observations and 34).
(JPG)

**S2 Fig. Plot of immunoprecipitated proteins belonging to ciliary and cytoskeleton Gene Ontology Cellular Component categories.** The axes show the log10(intensity) and log2 (intensity ratio between knock-in and control samples). Each dot corresponds to one detected protein; shape (circle, square, triangle) represents experimental batch; the colors summarize the number of times the enrichment ratio was above 2 for the protein: 0-orange/red; 1-green; 2-blue;3-purple. Text identifies proteins that had enrichment ratios >2 in each of the three experiments. The panels show proteins reported in cilia (left) or cytoskeleton (right).
(JPG)

**S3 Fig. Map of position of PC1 and NNT peptides onto their respective cognate proteins identified in IP eluents.** The data for each of the three experiments are shown. eGFP was also identified in each of the three experiments but not shown here or included in S1 Table since it is not part of the native protein nor is it an independent interactor of PC1.
(JPG)

**S1 Raw images. Original uncropped images of western blots shown in Fig 1.** This figure also includes in Fig 1G (on left) uncropped images of protein lysates isolated from Murine Embryonic Fibroblasts (MEFS) and probed with 7e12 antibody to test for the monoclonal's specificity. "WT" indicates lysate from a normal control mouse while"KO" identifies lysates of MEFs from a *Pkd1* null mouse [25]. PC1 is only detected in the control sample. The right panel is of lysates from the same samples shown on the left probed with an anti-HA monoclonal. Different blots were used for the two panels because both primary antibodies are of mouse origin. Anti-HA detects multiple non-specific bands in both kidney and MEFs.
(PDF)

**S1 Table. Table of all proteins identified in each of the three experiments.**
(XLSX)

**S2 Table. PC1 Extended Interactome and their subcellular localization.**
(XLSX)

**S3 Table. Pathway analyses of the PC1-interactome.**
(XLSX)

**S4 Table. Kidney and body weights supporting Fig 7A.**
(XLSX)

**S5 Table. Kidney and body weights supporting Fig 7B.**
(XLS)

**S6 Table. Proteins previously reported as PC1-interactors.**
(XLSX)

## Acknowledgments

We thank Jeff Reece, MS, Director of the Advanced Light Microscopy & Image Analysis Core, NIDDK and Dr. Jiji Chen at the Advanced Light Microscopy and Image Analysis Core, National Institute of Biomedical Imaging and Bioengineering (NIBIB) for helpful suggestions for imaging studies; Dr. Chengyu Liu, Director of the Transgenic Core Facility at the National Heart, Lung, and Blood Institute (NHLBI) for help developing the CRISPR/Cas9 knock-in model; Dr. Zu-Xi Yu at the Pathology Facility, NHLBI, for help histological specimens. This work utilized the computational resources of the NIH HPC Biowulf cluster (http://hpc.nih.gov). Cartoons in Figs 1 and 4 were generated using BioRender.com; Fig 4A was adapted from "Protein Peptide Mass Spectrometry", retrieved from https://app.biorender.com/biorender-templates.

## Author Contributions

**Conceptualization:** Cheng-Chao Lin, Luis F. Menezes, Gregory G. Germino.

**Data curation:** Cheng-Chao Lin, Luis F. Menezes.

**Formal analysis:** Luis F. Menezes.

**Funding acquisition:** Gregory G. Germino.

**Investigation:** Cheng-Chao Lin, Luis F. Menezes, Jiahe Qiu, Elisabeth Pearson, Fang Zhou, Yu Ishimoto.

**Methodology:** Cheng-Chao Lin, D. Eric Anderson.

**Supervision:** Luis F. Menezes, Gregory G. Germino.

**Validation:** Jiahe Qiu, Elisabeth Pearson.

**Writing – original draft:** Luis F. Menezes, Gregory G. Germino.

**Writing – review & editing:** Cheng-Chao Lin, Luis F. Menezes, Gregory G. Germino.

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
