## [Decision Letter · Decision Letter 0]

2 May 2023

PONE-D-23-07776In vivo Polycystin-1 interactome using a novel Pkd1 knock-in mouse modelPLOS ONE

Dear Dr. Germino,

Thank you for submitting your manuscript to PLOS ONE. After careful consideration, we feel that it has merit but does not fully meet PLOS ONE’s publication criteria as it currently stands. Therefore, we invite you to submit a revised version of the manuscript that addresses the points raised during the review process. Specifically, please address General Questions from Reviewer #2.

We look forward to receiving your revised manuscript.

Kind regards,

Weining Lu, MD

Academic Editor

PLOS ONE

Journal Requirements:

"This research was supported by the NIH, National Institute of Diabetes and Digestive and Kidney Diseases (NIDDK) Intramural Research Program, grant 1ZIADK075042. "   

7. PLOS ONE now requires that authors provide the original uncropped and unadjusted images underlying all blot or gel results reported in a submission’s figures or Supporting Information files. This policy and the journal’s other requirements for blot/gel reporting and figure preparation are described in detail at https://journals.plos.org/plosone/s/figures#loc-blot-and-gel-reporting-requirements and https://journals.plos.org/plosone/s/figures#loc-preparing-figures-from-image-files. When you submit your revised manuscript, please ensure that your figures adhere fully to these guidelines and provide the original underlying images for all blot or gel data reported in your submission. See the following link for instructions on providing the original image data: https://journals.plos.org/plosone/s/figures#loc-original-images-for-blots-and-gels.

In your cover letter, please note whether your blot/gel image data are in Supporting Information or posted at a public data repository, provide the repository URL if relevant, and provide specific details as to which raw blot/gel images, if any, are not available. Email us at plosone@plos.org if you have any questions

Reviewers' comments:

Reviewer's Responses to Questions

**Comments to the Author**

1. Is the manuscript technically sound, and do the data support the conclusions?

Reviewer #1: Yes

Reviewer #2: Yes

2. Has the statistical analysis been performed appropriately and rigorously? 

Reviewer #1: Yes

Reviewer #2: Yes

3. Have the authors made all data underlying the findings in their manuscript fully available?

Reviewer #1: Yes

Reviewer #2: Yes

4. Is the manuscript presented in an intelligible fashion and written in standard English?

Reviewer #1: Yes

Reviewer #2: Yes

5. Review Comments to the Author

Reviewer #1: This is a novel, rigorous study from an outstanding laboratory with extensive experience in the PKD field. The authors have established a new Pkd1 mouse model that employed the CRISPR/Cas9 system to insert an eGFP protein plus triple HA tag at the C-terminus of the endogenous mouse Pkd1 gene. Although the authors did not fully achieve their goal to generate an easily detectable fluorescent PC1-eGFP fusion to visualize the native GFP tagged protein, they were able to perform a novel IP approach to identify in vivo PC1-interacting proteins. They used a rational approach to limit the confirmed interactors to those identified in at least two of three experiments. There was enrichment for mitochondrial proteins and metabolic pathway components, such as NNT, which are important, novel observations.

Reviewer #2: The manuscript by Lin et al describes the generation of a new mouse model in which a GFP-3XHA tag was inserted into the end of the terminal coding exon of Pkd1. The purpose of this mouse model was for biochemical analysis (processing, interactions) and visualization of the endogenous polycystin-1 protein. Analysis of polycystin-1 is notoriously difficult due to its low-level expression and the large size of the protein along with the poor specificity and avidity of the existing antibody resources. Additionally, there is a large amount of conflicting data in the literature into the functions and localization of the polycystin-1 protein and this remains a major challenge for the field. In part this may be due to the dependency to use over expression systems to analyze the protein. Thus, the new Pkd1-GFP-3XHA model would provide an important experimental system in which to characterize this protein and its network.

The authors show that the fusion protein is fully functional as no cysts are evident in the Pkd1-GFP-3XHA mouse. In this study, they attempted to visualize the endogenous protein as well as conduct affinity mass spectrometry to explore polycystin-1 protein interactions. Unfortunately, imaging using the GFP was not successful without antibody amplification using GFP antibodies, possibly due to level of expression of the endogenous protein. They show using antiGFP that the fusion protein is in the cilium, but do not indicate if any junctional localization or other known sites for polycystin-1 were observed.

They utilize the new mouse model to perform a low stringency interactome study to detect weak interactions using P1 mouse brain tissue. They focused on interactions detected in at least 2 out of the 3 experiments. They report enrichment for proteins in the mitochondria and with metabolism (consistent with the PKD cellular phenotype) and importantly, also that they found the strongest interaction with polycystin-2 that is a well established polycystin-1 binding partner. They also detected an interaction with a recently published interactor Nnt, which was confirmed by IP western analysis. My concern with their approach is the level of stringency may be too lenient as both experimental and control IPs identified similar proteins, although their data does show enrichment for some networks relative to the background. They then use these data to establish a large STRING interactome network; however, based on the low stringency it also raises concerns about the strength of these networks. Curiously, their enrichment did not include ciliary proteins where they show the polycystin-1-GFP-3HA protein localizes. Also of the 84 published polycystin-1 interactors, the authors report enrichment of 6-7 proteins in their IPs, but only polycystin-2 was included in more than 2 of their IP studies.

Overall, this is an important mouse model that will provide additional opportunities to analyze endogenous polycystin-1; however, not in live cells and tissues as hoped. They demonstrate it can be immunoprecipitated and that they can detect proteins known to interact with polycystin-1. In my opinion, the current study design may have some limitations with regard to the protein network analysis and the stringency with which the study was performed. With that said, it would be good to have this model and the data reported and it would be of interest to many PKD investigators.

General questions:

Figure 1G: Western blots should have a loading control to ensure that P8 sample is not simply a poorer quality lysate. Also in these westerns, it would be beneficial to include a Pkd1 conditional null lysate to confirm specificity when using Pkd1 antibodies.

Figure 1H and 1I: Can the authors indicate that the size of the cleaved product is correct with regards to the inclusion of the GFP-HA amino acids? These seem to be small as GFP itself would be ~30kdal.

Figure 2E: Do Pkd1 ko heterozygous mice ever develop liver cysts that could explain the phenotype in this one mouse?

Figure 3B: (figure legend should read) …ciliary basal bodies…

Figure 3: Do the authors observe the same staining pattern with HA antibodies?

Line 370 should indicate figure 5A, not 4A, for the interactome

Line 375 should indicate figure 5B (also other call outs in that section)

Figure 8: indicates 7 known interactors were in the authors polycystin-2 interactome student but Venn diagram shows 6? Is the other the Xenopus protein?

6. PLOS authors have the option to publish the peer review history of their article (what does this mean?). If published, this will include your full peer review and any attached files.

Reviewer #1: No

Reviewer #2: No

---

## [Author Response · Author response to Decision Letter 0]

10 Jul 2023

July 6, 2023

RE: Manuscript PONE-D-23-07776

Dear Dr. Lu

Thank you for your note and the thoughtful reviews of our paper. Accompanying this cover letter is a revised version of our manuscript that addresses the questions posed by reviewer 2 in both a clean and “track changes” version, and which complies with the instructions in your note. 

Reviewer #1: 

We thank them for their very positive review. We note that there were no questions or issues raised by this reviewer for us to address. 

Reviewer #2:

We appreciate the reviewer’s generally positive comments about the study and welcome the opportunity to address questions/issues that they raised. I will address them in order of their presentation in the reviewer’s comments (red text below):

1. “They show using antiGFP that the fusion protein is in the cilium, but do not indicate if any junctional localization or other known sites for polycystin-1 were observed.” We also see an intracellular pattern which we think is in mitochondria but the signal is barely above baseline and the pattern is very similar to what we see in control cells, so we are not confident it is our tagged PC1. We do not see signal above background at cell-cell or cell-matrix sites. 

2. “My concern with their approach is the level of stringency may be too lenient as both experimental and control IPs identified similar proteins, although their data does show enrichment for some networks relative to the background. Curiously, their enrichment did not include ciliary proteins where they show the polycystin-1-GFP-3HA protein localizes. Also of the 84 published polycystin-1 interactors, the authors report enrichment of 6-7 proteins in their IPs, but only polycystin-2 was included in more than 2 of their IP studies… In my opinion, the current study design may have some limitations with regard to the protein network analysis and the stringency with which the study was performed.” We acknowledge the limitation detected by the reviewer. In fact, our discussion mentions the possibility that some of the interactors could be false positives. However, our rationale for this analysis was not unprecedented. Some groups reported similar thresholds when investigating interactors (ratio > 1.3 in [1]; ratio > 1.48 in [2]; ratio > 2 in [3]; ratio > 1.4 in [4]). As this short list suggests, choosing thresholds can be somewhat arbitrary. One approach that had been used to identify strong, direct binding partners is to use Grubbs test for outliers [2]. Using this strategy in our dataset, only PC1, PC2 and one immunoglobulin (likely because this is an immunoprecipitation experiment) are consistently detected in all experiments. So, as we mention in the discussion, it is possible that only PC2 is a PC1 direct interactor. However, we hypothesized that the inclusion of transient and indirect binding partners would be informative. To obtain such a list, we excluded all proteins that had a ratio of less than 1 in at least one of the experiments, and we included proteins that had ratio of at least 2 in at least two experiments. The rationale was that this strategy would enrich for proteins that were consistently more likely to be immunoprecipitated with PC1. This yielded our “extended interactome” of >800 proteins. For comparison, similar criteria, if applied to the control group (consistently more likely to be immunoprecipitated in the control group) yields only 4 proteins. This suggested to us that, while there were likely to be false positives, network patterns of this interactome could reflect PC1 functional roles. Our results corroborated this hypothesis, as one of the binding partners was validated and the network analyses was consistent with functional roles PC1 is believed to have. As the reviewer also correctly pointed out, however, the small overlap with previously published lists of PC1 interactors was puzzling, particularly given the nature of our experiment with less stringent washes and arguably lenient inclusion criteria. As we mention in the discussion, perhaps this discrepancy reflects differences in methods [4], tissue source, or using endogenous rather than over-expression of recombinant PC1 as bait. 

We also agree that is it interesting that there wasn’t relative enrichment of ciliary proteins, though we note that some ciliary proteins were identified in the extended set. There are multiple possible explanations for their low abundance: a) as a percentage of PC1 in the cell, that within the cilium comprises a very small fraction. While this may seem paradoxical given that the only place we see it unambiguously is in the primary cilium, the volume of the structure is extremely small relative to that of the rest of the cell, and if PC1 is distributed widely within the cell, as is suspected, rather than concentrated as it in primary cilia, it could explain the discrepancy in visualization; b) it is possible that ciliary interactions are less stable; c) permeabilization methods either disrupted interactions or failed to solubilize some membrane complexes. 

General questions:

1. “Figure 1G: Western blots should have a loading control to ensure that P8 sample is not simply a poorer quality lysate. Also in these westerns, it would be beneficial to include a Pkd1 conditional null lysate to confirm specificity when using Pkd1 antibodies.” We have repeated the experiment with new samples and a loading control to confirm that the measured amounts that were loaded were in fact similar and transferred correctly. To our surprise, and in contrast to what we had seen earlier and what had been previously reported by others [5], we did not see a decrease in PC1 expression in mouse kidneys from P1 to P8. We repeated the experiment multiple times to be sure there wasn’t an intermittent problem with transfer of high molecular weight proteins, but we got the same result: PC1 expression did not drop. We then generated another set of mice, prepared fresh samples, and again repeated the study. The results were similar---we saw no drop-off in PC1 expression. Looking at all of the data (ours and published literature), it seems there may be some variability in this pattern, but in light of our most recent data we have decided to stay silent on the issue in this revision. In either case, it doesn’t change any of our studies’ broad conclusions or implications. 

With respect to the reviewer’s other point, we unfortunately cannot do the requested study because the Pkd1 null state results in embryonic lethality, and none of the Cre recombinases available to us completely delete Pkd1 in the kidney. We have instead redone the experiment using murine embryonic fibroblasts from control and Pkd1 null mice, and have now included those data supplementary file S1-raw-images.

2. “Figure 1H and 1I: Can the authors indicate that the size of the cleaved product is correct with regards to the inclusion of the GFP-HA amino acids? These seem to be small as GFP itself would be ~30kDa.” In our prior studies of cultured cells, we saw predominantly two cleaved products of the correct predicted size (with HA tag) and then one smaller product at variable levels of about 8kDa which we interpreted to be a degradation product [6]. In this study, we also see a product of ~40-50kDa but the smaller products are predominant (eGFP + ~8kDa). We also see variable amounts of CTF and P100, other well-described PC1 cleavage products. Whether these differences reflect in vivo pattern differences or are the result of degradation during harvesting and processing the protein lysates is unclear. 

3. “Figure 2E: Do Pkd1 ko heterozygous mice ever develop liver cysts that could explain the phenotype in this one mouse?” No, we do not observe this. We think PC1 levels are generally just above threshold necessary to maintain tubular integrity, and for whatever reasons, the tag in this mouse was processed less well and resulted in a reduction of PC1 activity below the threshold [7]. 

4. “Figure 3B: (figure legend should read) …ciliary basal bodies…”. Corrected. Thank you.

5. “Figure 3: Do the authors observe the same staining pattern with HA antibodies?” Yes, but the signal is less bright and there is more background. We have included a representative image for the reviewer (Fig R1), but given the high background and limited additional information, we would prefer not to include this in the paper. 

6. “Line 370 should indicate figure 5A, not 4A, for the interactome.” Thank you. We have made the correction. 

7. “Line 375 should indicate figure 5B (also other call outs in that section).” Our apologies. We have made the correction. 

8. “Figure 8: indicates 7 known interactors were in the authors polycystin-2 interactome student but Venn diagram shows 6? Is the other the Xenopus protein?” Yes, and we have now explicitly stated that in the manuscript. 

In closing, we again thank the reviewers for their careful read of the manuscript and their suggestions for improvements. We hope you agree that we have satisfactorily addressed the questions and agree with the reviewers that “it would be good to have this model and the data reported, and it would be of interest to many PKD investigators.”

Thank you for your careful consideration.

Gregory Germino

National Institute of Diabetes and Digestive and Kidney Disease

REFERENCES

1. Blagoev B, Kratchmarova I, Ong SE, Nielsen M, Foster LJ, Mann M. A proteomics strategy to elucidate functional protein-protein interactions applied to EGF signaling. Nat Biotechnol. 2003;21(3):315-8.

2. Selbach M, Mann M. Protein interaction screening by quantitative immunoprecipitation combined with knockdown (QUICK). Nat Methods. 2006;3(12):981-3.

3. Hubner NC, Bird AW, Cox J, Splettstoesser B, Bandilla P, Poser I, et al. Quantitative proteomics combined with BAC TransgeneOmics reveals in vivo protein interactions. J Cell Biol. 2010;189(4):739-54.

4. Wang T, Gu S, Ronni T, Du YC, Chen X. In vivo dual-tagging proteomic approach in studying signaling pathways in immune response. J Proteome Res. 2005;4(3):941-9.

5. Wodarczyk C, Rowe I, Chiaravalli M, Pema M, Qian F, Boletta A. A novel mouse model reveals that polycystin-1 deficiency in ependyma and choroid plexus results in dysfunctional cilia and hydrocephalus. PLoS One. 2009;4(9):e7137.

6. Lin CC, Kurashige M, Liu Y, Terabayashi T, Ishimoto Y, Wang T, et al. A cleavage product of Polycystin-1 is a mitochondrial matrix protein that affects mitochondria morphology and function when heterologously expressed. Sci Rep. 2018;8(1):2743.

7. Qiu J, Germino GG, Menezes LF. Mechanisms of Cyst Development in Polycystic Kidney Disease. Adv Kidney Dis Health. 2023;30(3):209-19.

---

## [Editor Report · Decision Letter 1]

26 Jul 2023

In vivo Polycystin-1 interactome using a novel Pkd1 knock-in mouse model

PONE-D-23-07776R1

Dear Dr. Germino,

We’re pleased to inform you that your manuscript has been judged scientifically suitable for publication and will be formally accepted for publication once it meets all outstanding technical requirements.

Kind regards,

Weining Lu, MD

Academic Editor

PLOS ONE
---

## [Editor Report · Acceptance letter]

28 Jul 2023

PONE-D-23-07776R1 

*In vivo* Polycystin-1 interactome using a novel *Pkd1* knock-in mouse model 

Dear Dr. Germino:

I'm pleased to inform you that your manuscript has been deemed suitable for publication in PLOS ONE. Congratulations! Your manuscript is now with our production department. 

Kind regards, 

on behalf of

Dr. Weining Lu 

Academic Editor

PLOS ONE